# Prevalence and Associated Factors of Sexual Victimization: Findings from a National Representative Sample of Belgian Adults Aged 16–69

**DOI:** 10.3390/ijerph18147360

**Published:** 2021-07-09

**Authors:** Evelyn Schapansky, Joke Depraetere, Ines Keygnaert, Christophe Vandeviver

**Affiliations:** 1Department of Criminology, Criminal Law and Social Law, Ghent University, 9000 Ghent, Belgium; Evelyn.Schapansky@UGent.be (E.S.); Joke.Depraetere@UGent.be (J.D.); 2Research Foundation—Flanders (FWO), 1000 Brussels, Belgium; 3Department of Public Health and Primary Care, Ghent University, 9000 Ghent, Belgium; Ines.Keygnaert@UGent.be

**Keywords:** sexual violence, sexually transgressive behavior, sexual aggression, sexual assault, individual-level correlates, risk factors, mental health, coercive strategies

## Abstract

Sexual victimization is a major public health, judicial, and societal concern worldwide. Nationally representative and comparable studies are still lacking. We applied a broad definition of sexual violence, including hands-off and hands-on victimization, and behaviorally specific questions to assess sexual victimization. Lifetime and 12-month prevalence estimates were obtained that are representative of the Belgian general population aged 16 to 69 with regard to sex and age. These estimates indicate that 64% experienced some form of sexual victimization in their lives, and 44% experienced some form of sexual victimization in the past 12 months. Logistic regression analysis revealed significant associations for sex, age, sexual orientation, the number of sexual partners, and the financial situation with sexual victimization. Furthermore, our data show that mental health is significantly worse in persons with a history of prior sexual victimization. Prevalence estimates for all forms of sexual victimization are presented and compared to other national and international studies on sexual victimization. This comparison suggests that prevalence rates may have been underestimated in extant research. The prevalence estimates obtained in this study demonstrate that all sexes and ages are affected by sexual victimization.

## 1. Introduction

Sexual victimization is a global problem with immediate and long-term consequences for an individual’s physical, sexual, social, and mental well-being [1]. To address this major public health problem, it is indispensable to understand its nature and extent. Drawing general conclusions about the scope of the problem is, however, difficult. Many studies on sexual victimization use convenience samples, mostly consisting of students e.g., [2], focus on female victims e.g., [3], thus excluding male sexual victimization [4], or apply different definitions of sexual violence [5], thus hampering cross-study comparisons. The resulting outcomes of those studies tend to differ considerably and are not generalizable on a national level which would, however, help to formulate policy recommendations and improve prevention approaches that do not focus on a specific group [4] but instead can be tailored to each group at risk. Therefore, nationally representative studies are needed [4]. This allows reliable information to be obtained on how many people in a population are affected by which forms of sexual violence and what their vulnerabilities are.

Prevalence rates in nationally representative studies that include both male and female sexual victimization exhibit considerable variability, ranging from 4.5 to 56% for women and 0.2% and 21% for men [6,7,8,9,10,11,12,13]. This variability in findings is largely due to differences in the conceptual and operational definitions of sexual violence which limits the comparability of existing studies and the ability to draw conclusions for prevention approaches [5,9,14,15]. Researchers apply different definitions of sexual violence, include different behaviors to study sexual victimization, and cover different time periods. In addition, researchers differ in their use of behaviorally specific questions (BSQs). BSQs describe an incident in concrete terms instead of using more general terms such as ‘rape’. Using BSQs in sexual violence research is strongly recommended and limits ambiguity in participant interpretation [3,4]. After all, ambiguous interpretations may lead to participants not recognizing sexual victimization experiences as such or not linking the questions to their own experiences or may result in similar incidents being interpreted differently by different participants. These might be reasons for the prevalence of sexual victimization being underestimated [3,4].

For example, the latest National Crime Victimization Survey [16] found that 0.17% of US residents experienced “any rape, attempted rape, or other type of sexual attack” in the past six months. Here, victimization was assessed without the use of BSQs, covered a period of six months, and included participants aged 12 years or older. The 2019 Crime Survey for England and Wales [17] found slightly higher rates. In the self-completion part of the survey, 2.9% of adults aged 16 to 59 reported that they had been a victim of (attempted) sexual assault in the past year. This survey uses BSQs and covers hands-off and hands-on behaviors. Hands-off behaviors describe acts that do not involve physical contact between perpetrator and victim (e.g., exhibitionism) while hands-on behaviors involve physical contact (e.g., touching or penetration of the body). Different to the US survey described above, it covers experiences since the age of 16 that occurred in the past year. Furthermore, rape was limited to penetration with a penis which is not in line with more recent definitions of rape and largely excludes women as potential perpetrators. The World Health Organization (WHO) [18], however, suggests a broad definition of sexual violence, including hands-off and hands-on behaviors, that does not specify the gender of the victim or the perpetrator:
Sexual violence is any sexual act that is perpetrated against someone’s will. It can be committed by any person regardless of their relationship to the victim, in any setting. It includes, but is not limited to, rape, attempted rape and sexual slavery, as well as unwanted touching, threatened sexual violence and verbal sexual harassment. (p. 3)

Focusing on Belgium, where the current study was conducted, only one nationally representative study is available which found that 5.6% of women and 0.8% of men were sexually victimized after the age of 18 [11]. The authors applied a narrow definition of hands-on sexual violence assessed without BSQs which can be regarded as a major limitation of this study. Another study representative for the Dutch-speaking part of Belgium assessed hands-off and hands-on sexual victimization before and after the age of 18 [19]. The authors found that 10.6% of women and 6.3% of men experienced some form of sexual victimization before the age of 18, and 17.4% of women and 2.3% of men after the age of 18. A study conducted in several European countries that used BSQs found higher prevalence rates in Belgium, namely, 20.4% for women and 10.1% for men between 18 and 27 years [20]. In the European comparison, the rates found in Belgium were the lowest.

Looking at neighboring countries of Belgium, one nationally representative study can be found. Haas et al. [10] defined sexual violence as violence and aggression that includes “being approached sexually in a way that is offensive, being touched against your will, being forced to do sexual things, or being forced to have sexual things done to yourself” [9] (p. 600). When generally asked whether they had ever experienced sexual violence, 34% of Dutch women and 6% of Dutch men reported that they were sexually victimized in their lifetimes. When lifetime victimization was assessed with more specific questions (e.g., “I was forced to perform or to allow oral sex.” (p. 601), 55.9% of women and 20.5% of men reported that they experienced at least one type of sexual victimization. This highlights again the importance of BSQs in sexual victimization research. Lower rates for women but similar rates for men were found in a large convenience sample of German students aged 19 to 31 (35.9% for women and 19.4% for men) [2].

This heterogeneity in assessment and the resulting discrepancies in prevalence rates hampers drawing strong conclusions about the magnitude of sexual victimization within and between countries [5,20]. Moreover, most studies presented above were conducted before the #MeToo movement emerged in 2017. This movement shifted the attention onto sexual violence and raised awareness of this issue among large parts of society [21,22]. For many, it has changed their understanding of what is considered sexually transgressive and has led some to realize that they had experienced sexual violence [22]. More recent, large-scale, and representative studies are therefore needed to grasp the full scope of the problem, increase the generalizability of the results, and formulate policy recommendations at a national level [4,23].

It is further crucial to tailor such recommendations to specific risk factors of certain groups within the general population. Knowledge about risk factors for sexual victimization in Belgium is, however, scarce. The studies presented above show that women generally have a higher risk of sexual victimization than men. Moreover, previous research shows that this risk seems to be higher among young people [7,19,24] and non-heterosexual people [25]. Other factors related to sexuality that are associated with sexual victimization are, among others, early sexual initiation and more lifetime sexual partners [26]. Moreover, sexual victimization may be more prevalent among people with a lower income [9].

Informed policy recommendations are needed because the effects of sexual victimization can be severe and long-lasting. These can have negative consequences on an individual, interpersonal, and societal level. Some victims of sexual violence may develop an acute stress disorder, post-traumatic stress disorder (PTSD), anxiety, depression, or substance abuse disorders which, in turn, may have social and economic consequences [27]. Moreover, people who were victimized are more likely to perpetrate sexual violence [28]. These outcomes are not only detrimental for the individual but also for society at large. Knowing how much victims are affected by these potential outcomes can facilitate changes in policy to reduce sexual victimization.

At a political level in Belgium, the current National Action Plan on violence advocates a holistic management of sexual violence victims. However, reliable figures on sexual violence in Belgium are missing. Hence the necessity to conduct a Belgian representative prevalence study examining all vulnerabilities to victimization that can be used to design preventive measures and policy guidelines tailored to the Belgian context. This study is not only relevant on a national level. We also provide comparisons with other European countries that analyze the methodological differences resulting in large discrepancies and poor comparability of prevalence rates.

More specifically, this study focuses on the magnitude of sexual violence and its associated factors in the Belgian general population aged 16 to 69. A separate study was conducted with Belgian residents aged 70 and older which used a different sampling scheme and face-to-face interviews instead of an online survey, see [29]. As such, we produce data that entails the lifetime and past-year prevalence of sexual victimization, representative with regard to sex and age across three age groups (16–24; 25–49; 50–69), factors associated with it that may be interpreted as individual risk factors (sociodemographic, socioeconomic, and sexuality-related factors) and individual outcomes (recent substance use and mental health indicators). To achieve this, we apply BSQs as well as a broad, inclusive definition of sexual violence to capture experiences regardless of gender and across a wide lifespan. We rely on well-established, existing measures which we have updated to incorporate evolutions in sexual violence research by, for example, incorporating the WHO [18] definition of sexual violence and avoiding a gender bias in item wording.

## 2. Materials and Methods

### 2.1. Sampling Procedure and Participants

This study utilized data collected between October 2019 and February 2020. The National Register, containing demographic information on all Belgian residents, was used as a sampling frame from which Belgian residents were sampled to participate in an online survey. A random disproportionate stratified sample consisting of an equal number of male and female participants equally divided into three age groups was drawn by the National Register. By applying a disproportionate stratified sample method, certain subgroups are overrepresented to ensure sufficient statistical power in each subgroup. To obtain representative estimates, this overrepresentation is corrected post hoc by applying survey weights (cf. infra).

In total, 20,760 Belgian residents between 16 and 69 years were contacted in three waves by post by the Belgian National Register in October and November 2019. The desired sample size was calculated using a margin of error of 2% and a significance level of 5%. The effect size was estimated at 10% based on previous research on prevalence rates for Belgium e.g., [20]. These sample size calculations resulted in an a priori sample size of *n* = 864 per subgroup leading to a total of 5190 participants. Taking non-response and refusals to participate into account, four times this number of participants were contacted (i.e., *N* = 20,760).

To limit self-selection bias, the study was presented as a survey about health, sexuality, and well-being. The respondents could access the self-administered online survey using either a Quick Response or QR code, that could be scanned using a smartphone, or a link indicated in the letter sent by the National Register. The survey was administered through the survey software Qualtrics (Qualtrics, Provo, UT, USA). Prior to participation, respondents were provided additional information on the study and an informed consent form. Only those who gave informed consent were able to participate in the survey. To increase response rates, participants received one reminder letter after two weeks and were informed about the possibility to take part in a lottery to receive a voucher worth 30 EUR. For the latter, they were redirected to a separate short questionnaire after completing the main survey to ensure that survey answers could not be linked to personal contact information.

Out of 2791 respondents who initiated the survey, 76% completed it. Respondents were excluded due to either not providing informed consent (*n* = 261), not completing the survey (*n* = 394), not meeting the age criteria for participation (i.e., between 16 and 69 years old; *n* = 5), completing the survey more than once (*n* = 15), and concerns about the quality of the responses (*n* = 1). This leaves a total final sample of *n* = 2115 and an overall response rate of 10.2%. This final sample allows calculating prevalence estimates with a margin of error of 2%. The margin of error in the six sex-by-age groups ranges from 3 to 6%.

Sociodemographic characteristics of the unweighted sample are summarized in Table 1. When comparing the highest level of education in this sample with that of the entire population, using publicly available population data, it can be seen that higher educated people are overrepresented in the current study. Almost half of all respondents (i.e., 46.7%) completed a level of higher education, while, on the population level, 36% of Belgian residents between 15 and 64 years completed a higher educational level [30]. The number of people that completed no or only primary education reflects the proportion on the population level (i.e., 7.6%) [30]. In our sample, 11% identified as non-heterosexual which is similar to rates obtained in other online surveys [31,32]. The researchers were provided with the distribution of men and women across age groups in the entire population aged 16 to 69 (Table 2). These comparisons show that young women are overrepresented in this sample. In the analysis of prevalence estimates, sample weights are used to correct for this bias in order to obtain representative estimates with respect to sex and age. A correction for other factors, such as education, was not possible because these numbers could not be obtained from the National Register. Due to changes in European Union (EU) law on data protection [33], sampling information drawn from the National Register could not be shared with the researchers. Thus, the sampling and contacting of the participants were executed by the National Register. We were, therefore, also limited in the number of reminders that could be sent to potential participants and were not able to perform a non-response analysis.

### 2.2. Measures

In order to measure sexual victimization, a scale was developed and translated into a total of five languages including the most frequently spoken by the research population at the time of the study (i.e., Dutch, French and English) and two additional languages (i.e., Arabic and Farsi). A face and content validity test was conducted among 37 participants of the target population (17 male and 20 female) and 36 experts (i.e., psychiatrists, psychologists, people working in LGBTQIA organizations, policymakers, and academics). The survey was adapted according to the feedback given by respondents and experts, was well-received, and considered as covering all relevant aspects.

#### 2.2.1. Assessment of Victimization

##### Lifetime Prevalence

A broad definition of sexual victimization, including hands-off and hands-on victimization, using behaviorally specific questions was applied. To assess respondents’ lifetime sexual victimization, 17 questions were asked. In order to assess overall lifetime victimization, we created a dichotomous variable out of all 17 items that indicated whether the respondent had experienced any of the 17 items or not. These items are based on the Sexual Experience Survey (SES)—Long Form Victimization [34,35], the National Intimate Partner and Sexual Violence Survey [36], and the Sexual Aggression and Victimization Scale [2]. The items were rephrased to avoid gender-binary questions. The lack of consent was rephrased according to the definition of sexual violence made by the WHO [18] (‘against my will’ instead of ‘without my consent’) and question-wording was modified to fit within the Belgian social and legal context. Overall, the 17 items can be grouped into hands-off (eight items, e.g., sexual comments, voyeurism, exhibitionism, distribution of sexual images) and hands-on sexual victimization (nine items). The latter being further grouped into sexual abuse (four items: unwanted kissing, fondling/rubbing, undressing, and touching during care) and attempted or completed rape (five items: (attempted) oral penetration, (attempted) vaginal/anal penetration, being forced to penetrate someone else). A list of these items can be found in Appendix A.

##### 12-Month Prevalence

Respondents were asked to indicate whether they had ever experienced this in their lives. If a respondent answered with ‘yes’, they were asked how many times this had happened in the past 12 months. For the first four items of hands-off sexual victimization, respondents could answer with never, less than monthly, monthly, weekly, or daily. For the remaining questions, a numeric answer was sought. The answers to the 12-month prevalence questions were recoded into a binary variable to reflect the absence (=never or 0) vs. presence (all other options) of victimization in the past 12 months.

### 2.3. Coercive Strategy

The hands-on victimization items were followed by a question regarding the type of coercion that was used. Respondents were asked which circumstances described best how the incident had happened. Items reflecting the coercion types of verbal pressure, (threat of) using force, and exploiting an incapacitated state were derived from the SES [34,35] and Krahé and Berger [2]. In these surveys, questions on the experience of sexual victimization include the coercive strategy used (e.g., “Has a man ever made (or tried to make) you have sexual contact with him against your will by threatening to use force or by harming you?”) [2] (p. 404). Our face validity test, however, showed that this format did not make it clear what to answer when the incident occurred under other circumstances. Therefore, we assessed the coercive strategy separately from the victimization type and allowed respondents to indicate that none of the coercion strategies apply. We further included the option that someone exploited their position of authority or power [37,38].

### 2.4. Assessment of Associated Factors

A number of demographical, socio-economical, and variables related to the respondents’ sexuality were included in the analyses. Next to respondents’ year of birth and their sex assigned at birth, participants were further asked about their highest completed level of education and current occupational situation. The variable age was computed by subtracting the participant’s year of birth from 2019, the year in which the sample was drawn. This was also applied to respondents who completed the survey in early 2020 as it was more likely that their date of birth was after the last recorded response in February.

The current financial situation was assessed by asking respondents whether, considering their monthly household income, they were able to make ends meet easily, fairly easily, with some difficulty, or with great difficulty. These four answer options were combined into a binary variable (easy vs. difficult). Furthermore, participants were asked about their sexual orientation (see Table 1) which was also grouped into a binary variable (heterosexual vs. non-heterosexual) for analysis.

Respondents were further asked whether they ever had sex in their lives, with sex being defined as oral, vaginal, or anal intercourse. Those who answered this question affirmatively were asked about their age at the first time they had sex (i.e., sexual initiation) and how many male and female sexual partners they had in their lives.

To reduce information loss due to missing values, age at sexual initiation was grouped into early vs. late sexual initiation, with 15 years or younger being considered early (cf. Epstein et al., 2018; Young et al., 2018). Respondents, all being at least 16 years old at the time of survey completion, who indicated that they never had sex were assigned to the category late initiation. The number of male and female sexual partners was combined into a total number. Respondents who indicated that they never had sexual intercourse were given the value 0.

#### Substance Use and Mental Health

Depressive symptoms were assessed with the Patient Health Questionnaire-9 (PHQ-9) [39] consisting of nine items to measure depression severity with a maximum sum score of 27. For the analysis in the current study, the cut-off score of 5, indicating a mild depression, was used to form a binary variable.

Anxiety was measured using the Generalized Anxiety Disorder-7 (GAD-7) scale [40]. The seven items are answered on a four-point scale ranging from 0 (=not at all) to 3 (=nearly every day). A binary variable was calculated based on the cut-off score of 10 suggesting a moderate GAD. Both these scales refer to the past two weeks.

PTSD symptoms were assessed using the PC-PTSD-5 scale [41]. Only those who had experienced at least one traumatic event were asked five screening questions that are answered with yes (=1) or no (=0). This scale refers to the past month and a cut-off score of 3 suggests PTSD. Those who never experienced a traumatic event were given the value 0.

To decrease the number of missing cases, person-mean imputation was applied to the PHQ-9 for those cases with less than two and the PC-PTSD-5 scale and GAD-7 with less than one missing value.

Furthermore, we assessed problematic drinking behavior with the Alcohol Use Disorders Identification Test-Concise (AUDIT-C) scale [42]. A sum score was calculated based on three questions that assess the frequency and number of alcoholic drinks typically consumed as well as how often more than six alcoholic drinks are consumed. A sum score of 5 for men and 4 for women suggests risky drinking behavior. Respondents who indicated they never drink alcoholic beverages were given the value 0.

Respondents were further asked whether they had ever consumed medication to sleep or calm down (i.e., sedatives), cannabis, or stimulants such as cocaine or amphetamines. Respondents could indicate that they had done so in the past 12 months, not in the past 12 months, or never. The first two answer options were aggregated to form a binary variable for further analysis.

### 2.5. Analysis

Analyses were conducted using R version 3.6.3 (R Foundation for Statistical Computing, Vienna, Austria). To adjust for oversampling and non-response, sample weights were computed based on the known population distribution of males and females in three age groups (see Table 2). These population proportions were provided by the National Register and reflect the Belgian population at the time of the sampling. The resulting prevalence estimates are therefore representative of the Belgian population aged 16 to 69 with regard to sex and age [43]. Sex and age differences, as well as the group differences of victims and non-victims regarding mental health, were tested on the weighted estimates using Chi-square and Fisher’s exact tests. Bonferroni corrections were applied to the pairwise comparisons of the three age groups. Effect sizes were calculated using the phi-coefficient (ϕ). A value of 0.1 indicates a small effect, 0.3 a medium effect, and 0.5 a large effect [44].

Multivariate logistic regression analyses were computed to examine the association of demographic (i.e., sex, age), and socioeconomic factors (i.e., financial situation, occupational status, and educational level) as well as sexuality-related variables with the likelihood of sexual victimization, both over the lifetime and in the past 12 months. All variables, without sample weights, were added simultaneously. Adjusted odds ratios are reported to indicate the risk of sexual victimization for that variable while adjusting for the effects of the other predictor variables in the model. The multicollinearity assumption of multivariate regression analyses was tested with the Variance Inflation Factor (VIF) and indicated no violation. The linearity assumption of continuous variables added to the analysis (i.e., number of sex partners) was tested with the Box-Tidwell test [45] which indicated a violation of this assumption. The variable was, therefore, dummy coded based on the median (i.e., 0–2 vs. >2 sex partners).

## 3. Results

### 3.1. Prevalence of Sexual Victimization

Overall, 64.1% (95% CI: 61.9–66.1) of Belgian residents between 16 and 69 years experienced some form of sexual victimization during their lifetimes, and 44.1% (95% CI: 41.9–46.2) in the past 12 months. More specifically, 59.3% (95% CI: 57.2–61.4) have experienced some form of hands-off victimization, and 30.4% (95% CI: 28.5–32.4) some form of hands-on victimization during their lifetimes. The detailed prevalence estimates for lifetime and past-year victimization, stratified by sex and age, are presented in detail in Table 3 and Table 4, respectively.

#### 3.1.1. Sex Differences

The lifetime prevalence of sexual victimization for women was 81% which is 1.7 times higher than the estimate for men (47.5%; X^2^ (2, *n* = 2117.2) = 256.6; *p* < 0.001; ϕ = 0.35). About four in five women (78%) and two in five men (41%) reported some form of hands-off sexual victimization. Hands-on victimization was reported by two in five women (42%) and one in five men (19%).

The difference in victimization rates between women and men in the past 12 months was smaller compared to lifetime prevalence rates but still significant (X^2^ (2, *n* = 2117.2) = 123.7; *p* < 0.001; ϕ = 0.24). More than half of women (55%) and a third of men (31%) experienced some form of hands-off sexual victimization in the past 12 months. Prevalence rates of hands-on victimization in the past 12 months were 10% and 6% for women and men, respectively.

#### 3.1.2. Age Differences

Lifetime prevalence rates show that, in general, respondents younger than 50 years were significantly more likely to report hands-off forms of sexual violence compared to those aged 50 to 69. The prevalence in the past 12 months for both men and women was highest in the youngest age group and lowest in the oldest for nearly all forms of sexual victimization. Women between 16 and 24 showed the highest prevalence (79%) and men between the ages of 50 and 69 the lowest (22.7%). A detailed overview of significant differences between the different age groups is shown in Table 5.

##### Hands-Off

Women under the age of 50 reported more hands-off victimization than women between 50 and 69 years, both over the life course and in the past 12 months. More specifically, the prevalence of display and distribution of sexual images was more than six times higher among women between 16 and 24 compared to those aged 50 to 69. The results for men were similar. Prevalence rates for both men and women between 16 and 24 did not differ significantly from those aged 25 to 49.

##### Hands-On

The age differences for hands-on sexual victimization were smaller among the female respondents compared to the hands-off forms. About 45% of women aged 16 to 24 and 40% of women aged 25 to 49 reported some form of hands-on victimization during their lifetime. No significant age differences were found for lifetime hands-on victimization among women. However, women younger than 50 years reported unwanted rubbing/fondling significantly more often than women aged 50 to 69. Women between 16 and 24 years reported unwanted touching during care more often than women between 50 and 69 years.

For men, the highest prevalence rates of hands-on victimization were found in the youngest age group while the oldest age group reported the lowest prevalence rates. A significant age difference was found for unwanted rubbing/fondling and overall hands-on sexual victimization in the past 12 months.

### 3.2. Coercive Strategies

Table 6 shows the coercion strategies used by the perpetrator for each type of victimization and aggregated for sexual abuse, rape, and any hands-on victimization. Strikingly, most respondents indicated that none of the provided coercion strategies applied. This is the case for all forms of sexual abuse, except for being undressed against one’s will, where the (threat of) using physical force was indicated most often. For completed rape, using physical force, or the threat thereof, was also indicated most often. Overall, the coercion strategy indicated most often was exploitation of the victim’s incapacitated state, at 25%. Looking at the coercion strategies disaggregated by sex, it becomes clear that women reported more often verbal pressure, threat or force, and that someone exploited their position of authority or power. Men and women reported equally often that their incapacitated state was exploited, while men indicated more often that none of the coercion strategies applied.

### 3.3. Associated Factors: Potential Risk Factors

The results of the logistic regression analysis and all adjusted odds ratios are summarized in Table 7.

#### 3.3.1. Sex and Age

All other factors being controlled for, women were more likely than men to be sexually victimized, both in their lifetimes and in the past 12 months. Comparing the different age groups, young adults were most likely to be victimized as compared to those aged 25 years or older and those between 25 and 49 years old were more likely than the oldest age group to be victimized, both in their lifetimes and in the past 12 months.

#### 3.3.2. Sexuality and Relationships

Seventy-eight percent (95% CI: 71.0–83.1) of non-heterosexual persons experienced some form of sexual victimization in their lifetimes and were about two times more likely to be victimized than heterosexual persons. About four in five of all respondents (82%, *n* = 1739) reported that they ever had sexual intercourse in their lives. For 15.3% (*n* = 318) of the entire sample, this occurred before the age of 16. This early sexual initiation was linked to an increased likelihood of sexual victimization. The association was, however, only significant for past-year victimization. Furthermore, half of the participants (*n* = 1043) indicated that they had three or more sexual partners in their lives which was associated with a higher likelihood of sexual victimization.

Almost half of all respondents (48.7%, *n* = 1030) reported to be living with a partner and 17.5% (*n* = 370) reported having a partner but not living together. Living with a partner was linked to a decreased risk of sexual victimization in the past 12 months when compared to respondents without a partner.

#### 3.3.3. Socio-Economic Factors

Respondents’ level of completed education was not significantly associated with the likelihood of sexual victimization. Also, being a student was not associated with an increased likelihood compared to (self-)employed people or voluntary workers. Inactive respondents, on the other hand, were less likely to be victimized. Furthermore, the logistic regression analysis revealed an increased likelihood of sexual victimization for those who described their financial situation as difficult.

### 3.4. Associated Factors: Substance Use & Mental Health

People who ever experienced hands-on sexual violence were significantly more likely to consume alcohol or other drugs than non-victims. The effects are, however, small. After applying sample weights, a problematic drinking behavior over the past month was reported by 42% of victimized persons compared to 37% of non-victimized persons (*X*^2^ (1, *n* = 2105.7) = 5.58, *p* = 0.018, ϕ = 0.05). Ever having used sleep medication or other sedatives was reported by 48% vs. 32% (*X*^2^ (1, *n* = 2105.2) = 47.26, *p* < 0.001, ϕ = 0.15), 32% vs. 23% ever having consumed cannabis (*X*^2^ (1, *n* = 2101.7) = 17.37, *p* < 0.001, ϕ = 0.09), and 9 vs. 5% ever having consumed stimulants (*X*^2^ (1, *n* = 2097.2) = 8.38, *p* = 0.004, ϕ = 0.06).

A comparison between these two groups on indicators of mental health showed that mental health was significantly worse in those who ever experienced hands-on sexual violence. Depressive symptoms over the past two weeks were reported by 55% of victims vs. 36% of non-victims (*X*^2^ (1, *n* = 2097.0) = 64.29, *p* < 0.001, ϕ = 0.18). Self-harming behavior over the lifespan was reported by 15% vs. 8% (*X*^2^ (1, *n* = 2108.3) = 27.06, *p* < 0.001, ϕ = 0.11), and having ever attempted to commit suicide by 9% vs. 4% (*X*^2^ (1, *n* = 2106.3) = 17.6, *p* < 0.001, ϕ = 0.09).

Furthermore, anxiety symptoms over the past two weeks were reported by 21% vs. 10% (*X*^2^ (1, *n* = 2093.4) = 43.06, *p* < 0.001, ϕ = 0.14), and post-traumatic stress symptoms over the past month by 19% vs. 8% (*X*^2^ (1, *n* = 2104.7) = 56.97, *p* < 0.001, ϕ = 0.16).

### 3.5. Comparison with Other Prevalence Studies

Table 8 provides a comparison of the current study’s prevalence estimates to those obtained by other prevalence studies. These studies were selected because they were conducted within the past ten years in Belgium [19,20] or in neighboring countries. The latter includes a representative Dutch study [10] and a non-representative but large-scale German study [2]. To match our prevalence estimates as much as possible with these prevalence rates, we adjusted our analysis regarding respondents’ age and victimization items. This comparison shows that our prevalence estimates are substantially higher than those obtained in the Netherlands [10] and former studies in Belgium [19,20]. Only the study conducted with a large student sample in Germany yielded similar results [2].

The study by Buysse et al. [19] is representative of the Dutch-speaking part of Belgium and included both hands-off and hands-on behaviors. However, the authors did not use BSQs and provide no overall lifetime prevalence rate which limits the comparability with other studies. The Dutch nationally representative study [10] also assessed hands-off and hands-on behaviors, but used at least in parts BSQs to do so, and found higher rates than Buysse et al. [19] in Flanders, Belgium. However, not all items were behaviorally specific (e.g., “I was raped.”, p. 600). As both studies included an age range similar to or larger than ours, we did not adjust the sample to match our results.

In a study with German students aged 19 to 31, Krahé and Berger [2] used BSQs but excluded hands-off victimization and incidents that occurred before the legal age of consent (i.e., 14 years in Germany). The authors also paired the questions with three coercion strategies, namely, verbal pressure, (threat of) physical force, and the exploitation of an incapacitated state. Consequently, experiences that did not involve one of those coercion strategies were not assessed. The same applies to the study by Krahé et al. [20] conducted in Belgium in which the coercion strategy that someone exploited their position of authority or power over the victim was added. Furthermore, the authors also assessed sexual victimization that occurred after the legal age of consent, which is 16 years in Belgium. For these two studies, we matched the age range of the respondents and included only hands-on victimization from which we excluded the item ‘Someone touched my intimate body parts during care’. In our data, it was not possible to exclude experiences that occurred before the legal age of consent to match the results further.

## 4. Discussion

### 4.1. Prevalence of Sexual Victimization

The current study provides prevalence rates of sexual victimization that are nationally representative with regard to sex and age for the Belgian general population aged 16 to 69. We further aimed to provide results that facilitate comparisons to victimization rates in other contexts. To do so, we provided estimates for every behavior separately as well as pooled estimates for different types of sexual victimization for the entire lifespan as well as the past year. By integrating well-established measures with recent developments regarding the scope and definition of sexual violence and applying BSQs, we were able to provide comparable estimates for future researchers.

Our findings show that sexual violence is prevalent in both women and men and in all age groups. Surprisingly, respondents between 16 and 24 years reported higher lifetime victimization rates than older age groups despite their shorter window of exposure to sexual violence. The difference was especially large for hands-off victimization and forms of it that may typically occur online, such as showing and distributing sexual images. Three potential explanations can be formulated for these findings.

First, an increase in technology-facilitated forms of sexual victimization may account for the higher lifetime prevalence in young adults. Research on technology-facilitated sexual victimization has shown that young adults aged 18 to 24 years are more likely to report lifetime experiences of technology-facilitated forms of sexual victimization [46]. In the current study, voyeurism and exhibitionism were the only types of hands-off victimization that were reported most often by women aged 50 to 69 years in their lives. This could indicate that these victimization forms are replaced by technology-facilitated behaviors among younger adults (e.g., sending sexual images instead of exposing oneself in person).

Second, younger people might have a higher level of awareness of sexually transgressive behaviors. Even though the BSQs used in this study prevent a potential bias in the interpretation of sexual victimization [4], experiencing a situation as occurring against one’s will might still be different depending on a person’s awareness of topics like consent and sexual violence. These topics have caught more attention in recent years due to the #MeToo movement, which was largely an online phenomenon that might have affected young people more [22]. In a study comparing the understanding of sexual consent across age groups, Graf and Johnson [47] found that young adults’ definition of consent more frequently reflected media campaigns and that they had a more detailed understanding of what consent entails. Furthermore, even though the views on sexuality as well as consent might be shifting in older generations [47], those views were more restrictive when those older adults were young. Therefore, a situation that would now be experienced as unwanted might have been an accepted behavior in their youth. As a consequence, the incident may not be remembered as sexually transgressive and is, therefore, not reported. This might especially be the case for less serious incidents which may also account for the lower hands-off victimization rates in adults aged 50 to 69 years found in this study. However, this difference in perception does not mean older adults donot experience sexual violence. In a recent Belgian study, one in twelve older adults (8.4%) indicated they were sexually victimized in the past 12 months [29].

Third, recall bias may also account for lower rates in older age groups. Older generations may have been sexually victimized in their youth but did not recall these experiences when they filled out the survey. For younger generations, on the other hand, less time passed between the unwanted sexual experience and the reporting thereof.

### 4.2. Associated Factors

We further provide an analysis of associated factors. Our findings suggest that being female, young, non-heterosexual, having a higher number of sexual partners and financial difficulties are associated with an increased likelihood of sexual victimization which is in line with past research [9,20,24,25,26]. The fact that the youngest age group in our study was most likely to be victimized in the past year is in line with past research that showed young adults and adolescents are especially at risk of sexual victimization [7,19].

Furthermore, health outcomes appear to be worse in victimized persons across several indicators of mental health and substance use. These rates are overall relatively high, not only among victims. Rates of depression and anxiety found in a nationally representative health survey in 2018 [48] are considerably lower than those found in the current study even though they were assessed with the same scales. A reason for this could be that the national health survey was conducted with interviews while we used a self-administered online survey. In our study, the feeling of anonymity was therefore likely higher whereas interviews may lead to more socially desirable answers.

### 4.3. Coercive Strategy

The coercion strategies indicated most often were the exploitation of the victim’s intoxicated state and the use or threat of physical force. However, most respondents, and especially men, who have been victimized indicated that none of the four given coercion strategies applied to their experience. This suggests that additional mechanisms may be underlying unwanted sexual behavior. Canan et al. [25] added three coercion tactics to the ones provided in the revised SES—Short Form Victimization [35]. One of these tactics, “Just doing the behavior without giving me a chance to say ‘no’ (e.g., surprising me with the behavior)”, was indicated most often in their study. Furthermore, the authors identified 12 additional types of perpetration tactics in participants’ open-ended narratives. This shows that coercive strategies in a legal sense do not fully cover the perpetration tactics used to impede the victim’s resistance. Future research should therefore apply a broader definition of coercion tactics to understand the dynamics underlying sexual victimization. This may be especially useful to better understand the dynamics involved in male victimization as men in our study indicated even more often than women that none of the provided coercion strategies applied.

### 4.4. Comparison with Other Prevalence Studies

In comparison to earlier prevalence studies, it becomes evident that our prevalence estimates are relatively high. The most comparable study in terms of victimization items that was also conducted in Belgium reports prevalence rates that are half as high as the ones from the current study [20]. One major difference to our study is that Krahé et al. [20] asked respondents to report sexual victimization that occurred since the age of consent (i.e., 16 years in Belgium), thus excluding child sexual abuse, whereas our prevalence estimates include victimization over the entire lifetime. The study conducted in Germany [2], on the other hand, assessed experiences since age 14, thereby including a larger window of exposure to sexual violence than the study conducted by Krahé et al. in Belgium [20]. Furthermore, these studies combined victimization items with three [2] or four [20] coercion strategies. If a respondent experienced some type of victimization but none of the coercive strategies applied, they could not answer that question affirmatively. In our face validity test of survey items, respondents stated that this answer format did not make it clear what to answer when the incident occurred under other circumstances [49]. Our results, however, support that the circumstances in which sexual violence occurs are more diverse, as proposed by Canan et al. [25]. These aspects might have led to an underestimation of sexual victimization in the study conducted by Krahé et al. [20]. Finally, we assessed hands-on victimization with nine different items, eight of which were included for the comparison of prevalence rates obtained by Krahé and Berger [2] and Krahé et al. [20] who assessed hands-on victimization with four different items. These four items asked about nonconsensual sexual touch, attempted and completed sexual intercourse, and other sexual acts. As it might not be clear for everyone what, for example, sexual touch entails, our approach was to specify the different behaviors that comprise hands-on victimization more by using BSQs. This might have resulted in higher prevalence rates because respondents’ memories were cued more toward specific incidents that they otherwise might not have recalled or not considered as falling into this category [4].

The study conducted by Buysse et al. [19] in the Dutch-speaking part of Belgium obtained much lower prevalence rates, especially for men. One reason for this could be a lower willingness to disclose victimization given that the study used telephone interviews. Furthermore, no BSQs were used to assess sexual victimization. The lower prevalence rates reported by Haas et al. [10] may also be explained by differences in item wording. While Haas et al. [10] used a large representative sample, not all items were behaviorally specific (e.g., “I was raped.”, p. 601). As highlighted earlier, using BSQs is crucial in order to obtain reliable responses in sexual victimization surveys [4].

All in all, these large discrepancies in prevalence rates of sexual victimization and in the way they are assessed highlight the need for comparable studies on sexual victimization. Future research should use BSQs to assess sexual victimization as well as assess and report prevalence rates in a way that is comparable. That way, cross-country comparisons of prevalence rates could become more suitable to actually explain potential differences in sexual victimization rates by differences in, for example, legislation instead of attributing them to differences in study designs.

### 4.5. Limitations

Like any self-report measure, this survey study might have been subject to recall bias. This bias was minimized by the use of BSQs which facilitate memory recall [10]. Nevertheless, this bias may account for the lower victimization rates in older respondents who are asked to recall events that may date back a long time. Furthermore, self-selection bias may have affected the results of this study. For example, respondents who experienced sexual violence might have been more likely to participate in this study. Self-selection bias was, however, minimized as much as possible by presenting the study as a study on health, sexuality, and well-being without emphasizing sexual violence. Moreover, prevailing taboos surrounding sexual violence, such as the myth that men cannot be victims [4], may have resulted in lower reporting among men. However, given the generally higher prevalence rates for men compared to previous research, this influence was fairly limited in our study. This may be in part due to the online survey design used which is associated with a higher sense of anonymity [50].

In addition, the response rate in this study of 10.2% is lower than expected. Initially, we planned to send out three reminders to the potential participants to increase response rates. Due to new data protection regulations [33] and requirements of the National Register, only one reminder was allowed which may have drastically impacted the response rates. Nonetheless, a lower response rate does not negate the representativeness of a sample. Cook et al. [51] state that sample representativeness is more important than sample size. As such, smaller samples can provide more reliable results compared to larger samples due to the representativeness of the sample. In our sample, younger women were overrepresented. Even though younger people may be more likely to participate in an online survey, an overrepresentation of the youngest age group was intentional given the disproportional stratified sample design. By using a probability sample and applying sample weights, we balanced out differences in response rates across subgroups and were able to provide estimates that are representative of the general population with regard to age and sex. In addition, the current sample size allowed for statements with a 2% margin of error. When focusing on the different sex-by-age groups, the margin of error increased to 3–6% which is still largely within the acceptable margin of error of 5% [52].

Another limitation is that we cannot distinguish between lifetime victimization during childhood and adulthood. Future studies should add a short follow-up question asking, for example, “Has this happened to you before you were 16, after, or both?”. This would further increase comparability and allow to analyze both child and adult sexual victimization as well as their relationship.

Furthermore, all results are derived from cross-sectional data, and we analyzed potential risk factors and potential negative health outcomes based on theoretical considerations and past research. We cannot say whether certain factors were present before or whether they are a consequence of the experiences reported and some factors, such as drinking behavior, may be risk factors and outcomes of sexual victimization at the same time [27].

Finally, based on the question wording in this study, it is not possible to determine to what extent the sexual victimization experiences were, in fact, technology-facilitated. Some forms, such as the forced display of intimate body parts, could have occurred face-to-face or online. Further research is needed to ascertain the prevalence of technology-facilitated forms of sexual victimization within different age groups.

## 5. Conclusions

The current study provides nationally representative prevalence estimates of hands-off and hands-on sexual victimization in the Belgian general population. Lifetime prevalence rates contribute to our understanding of the magnitude of sexual victimization, and 12-month prevalence rates allow us to provide data on the current figures on sexual victimization to formulate policy recommendations. Our study shows that the extent of sexual victimization in the general population has been underestimated so far. Our risk analysis shows that young people, especially women, and non-heterosexual people are most at risk of sexual victimization. Nonetheless, the generally high prevalence rates found in our study demonstrate that no population group should be excluded from measures taken to prevent sexual victimization and the negative consequences that may follow. Prevention strategies should rather take the specific risk factors relevant for each group into account. Further research is needed to determine which of the potential risk factors should be targeted in a given subpopulation. Considering how widespread sexual violence is, it is necessary that public institutions have prevention and intervention strategies in place that consider and address victims of all ages and sexes to facilitate appropriate care. Our study has shown that there is a strong need to tackle the issue of sexual violence because a majority of society is affected by it and sexual victimization appears to be a predictor for further detrimental outcomes that require attention and prevention. Furthermore, in our study, sexual victimization was linked to financial difficulties. Taken together, this indicates a strong need to provide better care for victims of sexual violence which involves better accessibility and affordability of care, for example, by reimbursing as many sessions as needed of evidence-based psychological care for victims.

## Figures and Tables

**Table 1 ijerph-18-07360-t001:** Sample composition (*n* = 2115), unweighted.

Variable		*n* (Valid %)
Sex ^1^	Female	1164 (55.0)
	Male	951 (45.0)
Age in years (*M* (SD), range)		35.84 (16.75), 16–69
Educational level	Primary education or none	155 (7.3)
	Secondary education	974 (46.1)
	Higher education	986 (46.6)
Occupational status	Active ^2^	996 (47.1)
	Student	693 (32.8)
	Inactive or other ^3^	426 (20.1)
Sexual orientation (*N* = 1997)	Heterosexual	1871 (89.2)
	Bisexual	95 (4.5)
	Homosexual	53 (2.5)
	Pan-, omnisexual	43 (2.0)
	Asexual	11 (0.5)
	Other	24 (1.2)

^1^ Defined as sex assigned at birth. When asked about their gender, 19 respondents (0.9%) indicated to define themselves as, for example, non-binary or trans or gave no answer. To reduce missing cases, we relied on sex assigned at birth for the purpose of this study. ^2^ Combines the following categories: Employed/independent, contributing family member, voluntary work. ^3^ Combines the following categories: Financial self-sufficiency or any other type of alternative choice of living, housewife/-man, not able to work because of ill health, on the job market/looking for a job, retired, other.

**Table 2 ijerph-18-07360-t002:** Sample weights.

Age Group	Sex	Population *N*	Population Proportion	Sample *n*	Sample Proportion	Population/Sample = Weights
Youngest: 16–24	Female	576,098	0.07	520	0.25	0.30
	Male	601,426	0.08	340	0.16	0.48
Middle: 25–49	Female	1,864,081	0.24	362	0.17	1.39
	Male	1,883,527	0.24	303	0.14	1.67
Oldest: 50–69	Female	1,475,820	0.19	282	0.13	1.41
	Male	1,458,421	0.19	308	0.15	1.27
Total		7,859,373	1.00	2115	1.00	1.09

**Table 3 ijerph-18-07360-t003:** Detailed and grouped weighted lifetime prevalence estimates sexual victimization, by sex and age.

	Men				Women			
Item	16–24 % (95% CI)	25–49 % (95% CI)	50–69 % (95% CI)	Total % (95% CI)	16–24 % (95% CI)	25–49 % (95% CI)	50–69 % (95% CI)	Total % (95% CI)
Staring	19.7 (14.1–26.8)	24.4 (20.8–28.5)	13.0 (9.9–16.8)	19.5 (17.2–22.0)	67.7 (59.7–74.8)	65.7 (61.4–69.8)	47.2 (42.2–52.2)	59.0 (56.0–62.0)
Comments	24.7 (18.4–32.2)	22.8 (19.2–26.7)	11.4 (8.5–15.0)	18.9 (16.6–21.4)	52.3 (44.2–60.3)	56.1 (51.6–60.5)	42.9 (38.0–47.9)	50.6 (47.5–53.6)
Showing images	24.4 (18.2–31.8)	16.2 (13.1–19.7)	7.8 (5.4–11.0)	14.4 (12.3–16.6)	31.3 (24.3–39.3)	23.2 (19.6–27.2)	8.9 (6.3–12.2)	19.0 (16.7–21.5)
Calls or texts	12.9 (8.4–19.3)	5.3 (3.6–7.7)	3.6 (2.0–6.1)	5.8 (4.5–7.5)	26.0 (19.4–33.7)	16.6 (13.5–20.2)	9.9 (7.3–13.4)	15.5 (13.4–17.8)
Voyeurism	3.2 (1.2–7.6)	4.0 (2.5–6.1)	0.3 (0.03–1.8)	2.5 (1.7–3.7)	2.7 (0.9–7.0)	3.3 (2.0–5.4)	3.2 (1.8–5.6)	3.2 (2.2–4.5)
Distributing images	2.3 (0.7–6.4)	1.0 (0.4–2.4)	0.6 (0.1–2.2)	1.1 (0.6–2.0)	3.5 (1.3–8.1)	1.4 (0.6–2.9)	0.0 (0.0–1.2)	1.2 (0.6–2.1)
Exhibitionism	8.8 (5.1–14.5)	5.9 (4.1–8.5)	3.2 (1.8–5.7)	5.4 (4.1–7.0)	20.2 (14.4–27.5)	19.3 (16.0–23.1)	26.2 (22.0–30.9)	22.1 (19.6–24.7)
Forcing to show body parts	4.4 (2.0–9.1)	3.6 (2.2–5.8)	2.9 (1.6–5.3)	3.5 (2.5–4.8)	14.2 (9.3–20.9)	6.9 (4.9–9.6)	2.5 (1.3–4.7)	6.3 (5.0–8.0)
**Any Hands–Off**	51.8 (43.9–59.6)	45.9 (41.5–50.3)	30.5 (26.0–35.4)	41.1 (38.1–44.2)	84.2 (77.3–89.4)	82.6 (78.9–85.7)	68.8 (63.9–73.3)	77.6 (75.0–80.1)
Kissing	12.4 (7.9–18.7)	13.9 (11.0–17.2)	8.1 (5.7–11.4)	11.5 (9.7–13.6)	20.0 (14.2–27.3)	20.2 (16.8–24.0)	24.1 (20.1–28.7)	21.6 (19.2–24.3)
Touching in care	5.3 (2.5–10.2)	4.3 (2.8–6.5)	3.9 (2.3–6.4)	4.3 (3.2–5.7)	10.6 (6.4–16.8)	9.1 (6.8–12.1)	11.3 (8.5–15.0)	10.2 (8.4–12.2)
Fondling/rubbing	9.1 (5.4–14.9)	7.9 (5.8–10.7)	6.5 (4.3–9.5)	7.6 (6.1–9.4)	26.5 (19.9–34.3)	23.2 (19.6–27.2)	22.7 (18.7–27.2)	23.5 (21.0–26.2)
Undressing	3.8 (1.6–8.3)	2.3 (1.2–4.1)	1.3 (0.5–3.2)	2.2 (1.4–3.3)	6.2 (3.1–11.5)	5.5 (3.8–8.0)	4.3 (2.6–6.9)	5.1 (3.9–6.7)
**Any Sexual Abuse**	22.7 (16.6–30.0)	19.5 (16.2–23.2)	14.3 (11.1–18.2)	18.0 (15.8–20.5)	41.2 (33.4–49.3)	37.3 (33.1–41.7)	38.0 (33.2–42.9)	38.1 (35.2–41.1)
Oral penetration	1.8 (0.4–5.6)	2.3 (1.3–4.1)	2.6 (1.3–4.9)	2.3 (1.5–3.5)	8.1 (4.5–13.8)	4.1 (2.6–6.4)	8.5 (6.0–11.8)	6.4 (5.0–8.1)
Attempt of oral penetration	4.4 (2.0–9.1)	1.3 (0.6–2.9)	0.6 (0.1–2.2)	1.5 (0.9–2.5)	8.3 (4.6–14.0)	5.5 (3.8–8.0)	4.3 (2.6–6.9)	5.5 (4.2–7.0)
Vaginal or anal penetration	0.9 (0.1–4.3)	1.3 (0.6–2.9)	0.0 (0.0–1.2)	0.8 (0.4–1.6)	8.7 (4.9–14.5)	6.1 (4.2–8.6)	8.9 (6.3–12.2)	7.5 (6.0–9.3)
Attempt of vaginal or anal penetration	1.5 (0.3–5.2)	1.0 (0.4–2.4)	0.6 (0.1–2.2)	0.9 (0.5–1.8)	5.6 (2.7–10.8)	5.2 (3.5–7.7)	6.0 (4.0–9.0)	5.6 (4.3–7.2)
Forcing to penetrate	1.8 (0.4–5.6)	1.0 (0.4–2.4)	0.6 (0.1–2.2)	1.0 (0.5–1.8)	0.4 (0.0–3.6)	2.5 (1.4–4.4)	0.7 (0.2–2.3)	1.5 (0.9–2.5)
**Any Rape**	5.9 (3.0–11.0)	5.3 (3.6–7.7)	4.2 (2.5–6.8)	5.0 (3.8–6.5)	19.6 (13.9–26.9)	13.5 (10.7–16.9)	17.0 (13.5–21.2)	15.7 (13.6–18.1)
**Any Hands–On**	24.4 (18.2–31.8)	20.1 (16.8–24.0)	15.6 (12.2–19.7)	19.1 (16.8–21.6)	45.2 (37.3–53.3)	40.0 (35.8–44.5)	42.6 (37.7–47.6)	41.7 (38.8–44.8)
**Any Sexual Victimization**	58.5 (50.5–66.1)	51.5 (47.0–55.9)	37.7 (32.9–42.7)	47.5 (44.4–50.5)	85.2 (78.4–90.2)	85.4 (81.9–88.3)	73.4 (68.7–77.6)	80.8 (78.3–83.1)

**Table 4 ijerph-18-07360-t004:** Detailed and grouped weighted 12-month prevalence estimates sexual victimization, by sex and age.

	Men				Women			
Item	16–24 % (95% CI)	25–49 % (95% CI)	50–69 % (95% CI)	Total % (95% CI)	16–24 % (95% CI)	25–49 % (95% CI)	50–69 % (95% CI)	Total % (95% CI)
Staring	18.3 (12.8–25.2)	15.5 (12.5–19.0)	8.8 (6.2–12.1)	13.4 (11.5–15.7)	63.7 (55.5–71.1)	50.0 (45.6–54.4)	25.9 (21.7–30.5)	42.9 (39.9–46.0)
Comments	22.4 (16.4–29.7)	16.5 (13.4–20.1)	7.8 (5.4–11.0)	14.2 (12.2–16.5)	46.2 (38.2–54.3)	42.0 (37.7–46.4)	24.8 (20.7–29.4)	36.1 (33.3–39.1)
Showing images	21.2 (15.4–28.4)	13.9 (11.0–17.2)	6.2 (4.1–9.1)	12.1 (10.3–14.3)	24.8 (18.4–32.5)	14.9 (12.0–18.4)	4.3 (2.6–6.9)	12.4 (10.5–14.5)
Calls or texts	11.8 (7.4–18.0)	4.6 (3.0–6.9)	3.2 (1.8–5.7)	5.2 (4.0–6.8)	19.2 (13.5–26.5)	9.1 (6.8–12.1)	3.2 (1.8–5.6)	8.4 (6.8–10.3)
Voyeurism	3.2 (1.2–7.6)	1.6 (0.8–3.3)	0.0 (0.0–1.2)	1.3 (0.7–2.2)	1.9 (0.5–6.0)	1.4 (0.6–2.9)	1.1 (0.4–2.8)	1.3 (0.8–2.3)
Distributing images	0.9 (0.1–4.3)	0.3 (0.0–1.5)	0.6 (0.1–2.2)	0.5 (0.2–1.2)	1.2 (0.2–4.8)	0.0 (0.0–0.9)	0.0 (0.0–1.2)	0.2 (0.0–0.7)
Exhibitionism	6.7 (3.6–12.0)	3.0 (1.7–5.0)	0.6 (0.1–2.2)	2.7 (1.8–3.9)	12.7 (8.1–19.2)	5.2 (3.5–7.7)	1.4 (0.6–3.3)	4.9 (3.7–6.4)
Forcing to show body parts	3.2 (1.2–7.6)	1.6 (0.8–3.3)	0.3 (0.0–1.8)	1.4 (0.8–2.4)	7.1 (3.8–12.7)	0.8 (0.3–2.2)	0.4 (0.0–1.8)	1.6 (0.9–2.6)
**Any Hands–Off**	46.2 (38.4–54.2)	33.0 (29.0–37.3)	20.8 (16.9–25.2)	30.5 (27.8–33.4)	78.1 (70.6–84.1)	62.2 (57.7–66.4)	38.0 (33.2–42.9)	55.4 (52.3–58.4)
Kissing	6.5 (3.4–11.7)	2.6 (1.5–4.6)	1.9 (0.9–4.0)	3.0 (2.1–4.2)	8.7 (4.9–14.5)	3.3 (2.0–5.4)	2.1 (1.0–4.2)	3.7 (2.6–5.0)
Touching in care	3.8 (1.6–8.3)	1.0 (0.4–2.4)	1.0 (0.3–2.7)	1.4 (0.8–2.4)	6.0 (3.0–11.2)	1.4 (0.6–2.9)	0.4 (0.0–1.8)	1.7 (1.0–2.7)
Fondling/rubbing	6.7 (3.6–12.0)	2.6 (1.5–4.6)	0.6 (0.1–2.2)	2.5 (1.7–3.7)	14.4 (9.5–21.2)	4.7 (3.1–7.0)	2.5 (1.3–4.7)	5.3 (4.1–6.9)
Undressing	2.9 (1.1–7.2)	0.3 (0.0–1.5)	0.3 (0.0–1.8)	0.7 (0.3–1.5)	1.7 (0.4–5.7)	0.3 (0.0–1.4)	0.0 (0.0–1.2)	0.4 (0.1–1.0)
**Any Sexual Abuse**	14.4 (9.6–20.9)	5.6 (3.8–8.1)	2.9 (1.6–5.3)	6.0 (4.7–7.6)	22.3 (16.2–29.8)	8.3 (6.1–11.1)	4.3 (2.6–6.9)	8.8 (7.2–10.8)
Oral penetration	1.5 (0.3–5.2)	1.3 (0.6–2.9)	0.0 (0.0–1.2)	0.9 (0.4–1.7)	2.3 (0.7–6.5)	0.6 (0.1–1.8)	0.4 (0.0–1.8)	0.7 (0.3–1.5)
Attempt of oral penetration	2.9 (1.1–7.2)	0.7 (0.2–2.0)	0.0 (0.0–1.2)	0.8 (0.4–1.6)	3.3 (1.2–7.8)	1.4 (0.6–2.9)	0.0 (0.0–1.2)	1.1 (0.6–2.0)
Vaginal or anal penetration	0.6 (0.0–3.9)	0.0 (0.0–0.9)	0.0 (0.0–1.2)	0.1 (0.0–0.6)	2.7 (0.9–7.0)	0.3 (0.0–1.4)	0.4 (0.0–1.8)	0.7 (0.3–1.4)
Attempt of vaginal or anal penetration	0.9 (0.1–4.3)	0.3 (0.0–1.5)	0.0 (0.0–1.2)	0.3 (0.1–0.9)	2.5 (0.8–6.8)	0.3 (0.0–1.4)	0.4 (0.0–1.8)	0.6 (0.3–1.4)
Forcing to penetrate	0.6 (0.0–3.9)	0.7 (0.2–2.0)	0.0 (0.0–1.2)	0.4 (0.1–1.1)	0.0 (0.0–3.0)	0.3 (0.0–1.4)	0.0 (0.0–1.2)	0.1 (0.0–0.7)
**Any Rape**	3.8 (1.6–8.3)	2.3 (1.2–4.1)	0.0 (0.0–1.2)	1.7 (1.0–2.7)	7.5 (4.1–13.1)	1.6 (0.8–3.3)	0.7 (0.2–2.3)	2.2 (1.4–3.3)
**Any Hands–On**	15.6 (10.5–22.3)	5.9 (4.1–8.5)	2.9 (1.6–5.3)	6.3 (5.0–8.0)	25.6 (19.1–33.3)	9.1 (6.8–12.1)	4.6 (2.8–7.3)	9.9 (8.2–11.8)
**Any Sexual Victimization**	49.7 (41.8–57.6)	33.7 (29.6–38.0)	22.7 (18.7–27.3)	32.1 (29.3–35.0)	79.0 (71.6–85.0)	62.7 (58.3–66.9)	39.0 (34.2–44.0)	56.2 (53.1–59.2)

**Table 5 ijerph-18-07360-t005:** Significant differences in the prevalence of forms of sexual victimization between age groups, weighted.

	Men (*n* = 1060)				Women *(n =* 1057)			
Items with Significant Age Differences	Effect Size ϕ	16–24 vs. 25–49	16–24 vs. 50–69	25–49 vs. 50–69	Effect Size ϕ	16–24 vs. 25–49	16–24 vs. 50–69	25–49 vs. 50–69
**Lifetime**								
Staring	0.13	n.s.	n.s.	**<0.001**	0.19	n.s.	**<0.001**	**<0.001**
Comments	0.15	n.s.	**<0.001**	**<0.001**	0.12	n.s.	n.s.	**<0.001**
Showing images	0.16	0.024	**<0.001**	**<0.001**	0.21	n.s.	**<0.001**	**<0.001**
Calls or texts	0.13	0.002	**<0.001**	n.s.	0.15	0.012	**<0.001**	0.005
Forcing to show body parts					0.16	0.007	**<0.001**	0.012
Any Hands–Off	0.17	n.s.	**<0.001**	**<0.001**	0.17	n.s.	**<0.001**	**<0.001**
Any Sexual Victimization	0.16	n.s.	**<0.001**	**<0.001**	0.15	n.s.	0.004	**<0.001**
**Past 12 months**								
Staring					0.28	0.004	**<0.001**	**<0.001**
Comments	0.15	n.s.	**<0.001**	**<0.001**	0.19	n.s.	**<0.001**	**<0.001**
Showing images	0.16	0.034	**<0.001**	**<0.001**	0.22	0.006	**<0.001**	**<0.001**
Calls or texts	0.13	0.002	**<0.001**	n.s.	0.19	<0.001	**<0.001**	**<0.001**
Exhibitionism	0.13	0.052	**<0.001**	0.025	0.17	0.003	**<0.001**	0.004
Forcing to show body parts					0.19	**<0.001**	**<0.001**	n.s.
Any Hands–Off	0.19	0.003	**<0.001**	**<0.001**	0.29	**<0.001**	**<0.001**	**<0.001**
Touching in care					0.14	0.003	**<0.001**	n.s.
Fondling/rubbing	0.13	0.028	**<0.001**	0.047	0.18	**<0.001**	**<0.001**	n.s.
Any Hands–On	0.17	**<0.001**	**<0.001**	0.048	0.23	**<0.001**	**<0.001**	0.013
Any Sexual Victimization	0.19	**<0.001**	**<0.001**	**<0.001**	0.29	**<0.001**	**<0.001**	**<0.001**

Abbreviations: n.s., not significant. Note. Displayed *p*-values refer to the uncorrected *p*-values. Significant *p*-values after Bonferroni correction are shown in bold (α_adjusted_ 0.05/60 = 0.00083).

**Table 6 ijerph-18-07360-t006:** Type of coercion used for sexual abuse, rape, and attempted rape, in %, overall and by sex.

Type of Victimization		Verbal		Force		Exploit		Authority		Other ^a^	
Kissing (*n* = 354)	Men (*n* = 109)	2.8	7.1	3.7	19.2	26.6	23.7	6.4	9.9	62.4	50.0
	Women (*n* = 245)	9.0		26.1		22.4		11.4		44.5	
Touching in care (*n* = 163)	Men (*n* = 43)	-	7.4	9.3	14.7	16.3	17.8	11.6	21.5	62.8	51.5
	Women (*n* = 120)	10.0		16.7		18.3		25.0		47.5	
Fondling (*n* = 361)	Men (*n* = 75)	-	7.2	4.0	13.9	13.3	18.0	13.3	15.0	72.0	57.1
	Women (*n* = 286)	9.1		16.4		19.2		15.4		53.1	
Undressing (*n* = 88)	Men (*n* = 24)	-	14.8	33.3	33.0	8.3	20.5	20.8	26.1	41.7	29.5
	Women (*n* = 64)	20.3		32.8		25.0		28.1		25.0	
**Any Sexual Abuse (*n* = 636)**	Men (*n* = 180)	1.7	8.2	7.8	18.4	22.8	22.6	10.0	15.9	66.7	59.3
	Women (*n* = 456)	10.7		22.6		22.6		18.2		56.4	
Oral penetration (*n* = 102)	Men (*n* = 21)	9.5	27.5	9.5	32.4	28.6	23.5	38.1	26.5	28.6	22.5
	Women (*n* = 81)	32.1		38.3		22.2		23.5		21.0	
Attempt of oral penetration (*n* = 96)	Men (*n* = 21)	19.0	27.1	28.6	27.1	28.6	18.8	19.0	22.9	33.3	28.1
	Women (*n* = 75)	29.3		26.7		16.0		24.0		26.7	
Vaginal or anal penetration (*n* = 99)	Men (*n* = 7)	14.3	26.3	57.1	30.3	42.9	27.3	28.6	25.3	-	24.2
	Women (*n* = 92)	27.2		28.3		26.1		25.0		26.1	
Attempt of vaginal or anal penetration (*n* = 75)	Men (*n* = 10)	-	12.0	20.0	20.0	10.0	22.7	-	16.0	70.0	45.3
	Women (*n* = 65)	13.8		20.0		24.6		18.5		41.5	
Forcing to penetrate (*n* = 24)	Men (*n* = 11)	45.5	45.8	-	16.7	45.5	29.2	9.1	25.0	9.1	16.7
	Women (*n* = 13)	46.2		30.8		15.4		38.5		23.1	
**Any Rape (*n* = 248)**	Men (*n* = 49)	18.4	25.8	20.4	26.2	26.5	24.6	24.5	23.0	40.8	33.5
	Women (*n* = 199)	27.6		27.6		24.1		22.6		31.7	
**Any Hands-On (*n* = 692)**	Men (*n* = 192)	6.2	12.9	9.4	21.1	24.5	25.0	12.0	17.9	66.1	60.0
	Women (*n* = 500)	15.4		25.6		25.2		20.2		57.6	

Note: Verbal = Verbal pressure; Force = Use or threat of using physical force; Exploit = Exploitation of an incapacitated state; Authority = Exploitation of a position of authority or power over the victim; Other = None of the above. ^a^ Respondents could provide multiple answers unless Other *=* None of the above was selected.

**Table 7 ijerph-18-07360-t007:** Logistic regression analysis of factors associated with sexual victimization.

Predictors	Lifetime aOR (95% CI)	Past 12 Months aOR (95% CI)
Sex (female)	4.96 (4.02–6.14) ***	3.43 (2.82–4.18) ***
Age		
16–24	2.13 (1.36–3.35) ***	3.52 (2.33–5.35) ***
25–49	1.56 (1.18–2.05) **	1.79 (1.37–2.34) ***
50–69	Ref	Ref
Sexual orientation (non-heterosexual)	1.83 (1.25–2.72) **	1.71 (1.23–2.39) **
Sexual initiation (early)	1.33 (0.96–1.85)	1.51 (1.13–2.03) **
Number of sexual partners (>2)	1.80 (1.42–2.30) ***	1.91 (1.52–2.42) ***
Relationship status		
No partner	Ref	Ref
Not living with partner	1.02 (0.74–1.40)	0.97 (0.73–1.30)
Living with partner	1.00 (0.73–1.35)	0.66 (0.50–0.88) **
Education level		
Primary or none	0.79 (0.51–1.23)	0.80 (0.53–1.22)
Secondary	1.15 (0.91–1.45)	1.19 (0.95–1.48)
Higher	Ref	Ref
Occupational status		
Inactive or other	0.72 (0.54–0.96) *	0.74 (0.55–0.98) *
Student	1.08 (0.70–1.65)	1.05 (0.71–1.56)
Active	Ref	Ref
Financial situation (difficult)	1.34 (1.06–1.71) *	1.30 (1.04–1.62) *

Abbreviations: aOR, adjusted odds ratio. *** *p* < 0.001, ** *p* < 0.01, * *p* < 0.05.

**Table 8 ijerph-18-07360-t008:** Comparison of prevalence studies on sexual lifetime victimization.

Author	Country	Sample		Prevalence Rates (%)		Items
		Age	*n*	Men	Women	
Buysse et al. [19]	BE (Flanders)	14–80	1825	<18y: 6.3% >18y: 2.3%	<18y: 10.6% >18y: 17.4%	Hands-off and hands-on, non-BSQ
Current study	BE	16–69	2117.2	47.5	80.8	Hands-off and hands-on, BSQ, entire lifespan
				(95% CI: 44.4–50.5)	(95% CI: 78.3–83.1)	
de Haas et al. [10]	NL	15–70	6428	20.5	55.9	Hands-off (limited) and hands-on, partially BSQ
Current study	BE	16–69	2117.2	47.5 (95% CI: 44.4–50.5)	80.8 (95% CI: 78.3–83.1)	Hands-off and hands-on, BSQ
Krahé and Berger [2]	GER	19–31	2149	19.4	35.9	Only hands-on, BSQ, based on SES, only experiences after age 14
Current study	BE	19–31	508.6	19.9 (95% CI: 15.1–25.6)	41.1 (95% CI: 35.2–47.2)	Only hands-on (without ‘touching in care’), BSQ, entire lifespan
Krahé et al. [20] Belgian subsample	BE	18–27	393	10.1	20.4	Only hands-on, BSQ, based on SES, only experiences after age 16
Current study	BE	18–27	368.8	20.5 (95% CI: 14.9–27.4)	41.6 (95% CI: 34.6–48.9)	Only hands-on (without ‘touching in care’), BSQ, entire lifespan

## Data Availability

The data is available upon request from the corresponding author.

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
