# Peer review of "Prevalence and Associated Factors of Sexual Victimization: Findings from a National Representative Sample of Belgian Adults Aged 16–69"

_ijerph, 2021, doi:10.3390/ijerph18147360_

Round 1
Reviewer 1 Report
Review of IJERPH-1279790
Prevalence and associated factors of sexual victimization: Findings from a national representative sample of Belgian adults aged 16-69.
June 23, 2021
This manuscript describes a well-conducted, crucially important study of sexual victimization prevalence and correlates in a nationally representative sample of Belgian adults. Given the paucity and significant limitations of research on this topic with this population, the present work is of vital importance both to the research field and more importantly to policy-making and intervention. The measures and methods used are both sound and consistent with those research practices that have the best support in the international literature. The results are largely consistent with extant results using similar methods, especially for younger and non-heterosexual respondents, lending them credence. The only thing I can find to comment on is the authors’ use of “sex” when they perhaps mean “gender”?
Author Response
Re: Revisions of the manuscript “Prevalence and Associated Factors of Sexual Victimization: Findings from a National Representative Sample of Belgian Adults Aged 16-69”
Dear editors, dear reviewers,
We would like to thank the reviewers very much for their valuable and constructive feedback. Their input has undoubtedly improved the quality of the article. To facilitate your evaluation of the reworked contribution, please find below a report of the revisions. This includes the reviewers’ comments followed by our response and a description of the changes made to the manuscript. The line numbers refer to the document with visible tracked changes.
Yours sincerely,
Prof. Dr. Christophe Vandeviver
Reviewer 1:
- “The only thing I can find to comment on is the authors’ use of “sex” when they perhaps mean “gender”?”
- We do, in fact, deliberately refer to sex instead of gender. To clarify this, a note was added to the category in Table 1 explaining that the variable ‘sex’ was defined as sex assigned at birth. We further added information on the variable gender in our study and why we decided to rely on sex assigned at birth instead (lines 211-213). Furthermore, it is mentioned in the methods section that respondents were asked about their sex assigned at birth (line 270).
Reviewer 2:
- Line 31 - I think there is a better opening sentence that gives some indication of the topic. Do not assume that every reader has read the abstract.
- An introduction sentence referring to a WHO report on the consequences of sexual victimization was added in lines 31 and 32.
- Line 40 - This may read better as "This allows reliable information to be obtained...", and
- Line 64 - I would suggest removing "next to" and just having it as "hands on behaviour"
- The sentences were adjusted accordingly.
- Line 133 - why have you not chosen to include over 69s? This could be explained further here.
- The population of Belgian residents aged 70 and older was subject of a separate study with a different research design and data collection method. We added a footnote to line 133 (now 143) to point this out.
Reviewer 3:
- The number reported in line 88 is high. Please add the sexual violence, is it based on the old WHO definition of the new one. This is the sentence: ... “found that 34% of Dutch women and 6% of Dutch men reported that they were sexually victimized in their lifetimes when generally asked whether they had ever experienced sexual violence” ... with which definition?
- The definition of sexual violence used by Haas et al. was integrated (lines 94-97).
- The same for line 90: "When lifetime victimization was assessed with more specific questions, 55.9% of women and 20.5% of men reported that they experienced at least one type of sexual victimization”.
- An example was added to illustrate the “more specific questions” used in their study (line 101).
- Lines 115-122, I think this paragraph is good for discussion or maybe conclusion; there is no actual policy recommendation in conclusion, a section like this can improve the decision. Also, can you add a discussion on the EU's law that restricted access to data by researchers? Considering the results of this study, one the strategy would be realizing data or even facilitating data not only for the researcher to do more work but also for the public to learn more about the areas with high prevalence of sexual violence?
- As this paragraph serves to highlight the relevance of the current study based on previous research, we decided not to move it to the discussion. Instead, specific policy recommendation, that follow from our study, were added in the conclusion (lines 678-680 and 683-687).
- Please define hands-on and hands-off in the earlier section (e.g., intro) and keep the current one in lines 223-232
- Hands-off and hands-on behaviors are now defined in the introduction when first referring to these categories (lines 67-70). Examples are given for further clarification.
- T5, edit n=1060.4 to n=1060 and 1056.8 to 1056
- 4 was rounded down to 1060 and 1056.8 was rounded up to 1057.
- Table 6 reported the modified version of some specific questions in the survey to make it comparable please report the findings based on age-cat, similar to T4.
- While we agree that a disaggregation by age category would improve comparability, we decided to only disaggregate Table 6 further by sex, but not by age. Splitting every cell up into three more cells would have resulted in multiple large tables that would be difficult for the readers to interpret. Furthermore, this would have partially resulted in very small cell counts whose relevance would be rather limited. We did, however, provide further details on sex differences, next to the overall percentages, which adds nuance to and improves comparability of the findings. Related to this, a short paragraph summarizing these differences was added to the results (lines 405-409) and an addition was made to the discussion (lines 566-567 and 577-579).
Additional significant changes:
- In Table 7, in case of binary variables, we exchanged the reference category with the category the adjusted odds ratio refers to as we believe that this allows better to correctly interpret the results depicted in Table 7.
- In the earlier version of the manuscript, the Appendix containing a list of the items used in this study (referred to in line 245) was missing. Appendix A can now be found at the end of the manuscript, before the reference list.
Reviewer 2 Report
Thank you for submitting this interesting paper on sexual victimisation in Belgian populations. Overall, I think this is a good paper with some very minor amendments required.
Introduction
This is a good introduction that makes use of wider literature and provides a rationale for the study.
Line 31 - I think there is a better opening sentence that gives some indication of the topic. Do not assume that every reader has read the abstract.
Line 40 - This may read better as "This allows reliable information to be obtained..."
Line 64 - I would suggest removing "next to" and just having it as "hands on behaviour"
Line 133 - why have you not chosen to include over 69s? This could be explained further here
Materials and Methods:
This section is very well explained. It is clear and justification has been made as to the methods used. Probably a comment for the Editors but having table 1 split over two pages as it is, does not make it very easy to read.
Results:
I find these really clear and easy to follow. They are explained well and presented appropriately.
Discussion:
The discussion is well considered. Wider evidence has been included to assist with the appraisal of the results. Key results are commented upon and differentiations between age groups has been considered. Recommendations for future research are made and exploration of the study's limitations is honest.
I have found this study were interesting and am not too surprised by the results but it is good to see them as evidence rather than assumption.
Author Response

(The authors gave the same response as above.)

Reviewer 3 Report
It is an exciting and most needed study, using an analytical sample of 2,115 participants between 16-69 years old in Belgium. The study used a very comprehensive survey that included two main types of sexual violence: hands-on and hands-off. The results reported sexual violence in the past 12 months and a lifetime. Overall, the study used appropriate techniques and tables to write the findings, here are a few suggestions to improve the quality of work:
Introduction:
1) The number reported in line 88 is high. Please add the sexual violence, is it based on the old WHO definition of the new one.
This is the sentence: ... “found that 34% of Dutch women and 6% of Dutch men reported that they were sexually victimized in their lifetimes when generally asked whether they had ever experienced sexual violence” ... with which definition?
The same for line 90: “
"When lifetime victimization was assessed with more specific questions, 55.9% of women and 20.5% of men reported that they experienced at least one type of sexual victimization”.
2) Lines 115-122, I think this paragraph is good for discussion or maybe conclusion; there is no actual policy recommendation in conclusion, a section like this can improve the decision. Also, can you add a discussion on the EU's law that restricted access to data by researchers? Considering the results of this study, one the strategy would be realizing data or even facilitating data not only for the researcher to do more work but also for the public to learn more about the areas with high prevalence of sexual violence?
3) Please define hands-on and hands-off in the earlier section (e.g., intro) and keep the current one in lines 223-232
4) T5, edit n=1060.4 to n=1060 and 1056.8 to 1056
5) Table 6 reported the modified version of some specific questions in the survey to make it comparable please report the findings based on age-cat, similar to T4.
Author Response

(The authors gave the same response as above.)
